# Leucoverdazyls as Novel Potent Inhibitors of Enterovirus Replication

**DOI:** 10.3390/pathogens13050410

**Published:** 2024-05-15

**Authors:** Alexandrina S. Volobueva, Tatyana G. Fedorchenko, Galina N. Lipunova, Marina S. Valova, Valeriya A. Sbarzaglia, Anna S. Gladkikh, Olga I. Kanaeva, Natalia A. Tolstykh, Andrey N. Gorshkov, Vladimir V. Zarubaev

**Affiliations:** 1St. Petersburg Pasteur Institute, 14 Mira St., St. Petersburg 197101, Russia; 2Postovsky Institute of Organic Synthesis, Ural Branch of the Russian Academy of Sciences, 22/20 S. Kovalevskoi St., Yekaterinburg 620108, Russia; 3Smorodintsev Influenza Research Institute, 15/17 Prof. Popova St., St. Petersburg 197376, Russia

**Keywords:** enteroviruses, coxsackievirus, leucoverdazyls, antioxidant, antiviral, 2C protein

## Abstract

Enteroviruses (EV) are important pathogens causing human disease with various clinical manifestations. To date, treatment of enteroviral infections is mainly supportive since no vaccination or antiviral drugs are approved for their prevention or treatment. Here, we describe the antiviral properties and mechanisms of action of leucoverdazyls—novel heterocyclic compounds with antioxidant potential. The lead compound, **1a**, demonstrated low cytotoxicity along with high antioxidant and virus-inhibiting activity. A viral strain resistant to **1a** was selected, and the development of resistance was shown to be accompanied by mutation of virus-specific non-structural protein 2C. This resistant virus had lower fitness when grown in cell culture. Taken together, our results demonstrate high antiviral potential of leucoverdazyls as novel inhibitors of enterovirus replication and support previous evidence of an important role of 2C proteins in EV replication.

## 1. Introduction

Enteroviruses (EVs) represent a diverse group of small icosahedral non-enveloped viruses with a single-stranded non-segmented positive RNA genome belonging to *Picornaviridae* family Enterovirus genus, encompassing EV A-L and rhinovirus A-C species [1]. EVs are characterized by high resistance to harsh environments and the ability to cause both self-limiting infections as well as life-threatening diseases and outbreaks, especially among newborns and children [2]. EV-A species members, such as coxsackieviruses (A6, A16) and enterovirus 71, are etiological agents of the largest outbreaks of hand, foot, and mouth disease (HFMD), both in Asia and Western countries. Enterovirus D68 (EV-D68) infection is associated with respiratory and neurologic disease worldwide. Although EVs are mainly associated with acute infections, more evidence is emerging on the long persistence of EVs in target organs, such as the heart and pancreas [3,4,5].

Among EV-induced diseases, HFMD is one of the most widely spread pathologies. It is mostly a mild self-limiting disease that occurs in children under the age of five. Its symptoms include sores in the mouth, anorexia, mild fever, as well as ulcers on the hands, feet, and mouth [6]. In some cases, however, EV infection (EVI) leads to fatal neurological or cardiopulmonary complications such as myocarditis, pulmonary edema, and neuroinflammation causing meningoencephalitis and cognitive impairment. EVI is also of high danger for children with immunodeficiencies or accompanying diseases. Therefore, HFMD is a significant concern for public health [7].

The vast majority of HFMD cases are caused by EV-A species enteroviruses, mainly EV-A71, CV-A16, CV-A10, and CV-A6. EV-B species also can cause sporadic cases of HFMD [8,9]. Enteroviruses are characterized by high genotypic and phenotypic diversity, which significantly complicates the development of broad-spectrum vaccines for the prevention of EVI. Currently, vaccination is available only for the prevention of poliomyelitis and for EV71-associated HFMD [10,11,12]. A large amount of research carried out in the field of the EV life cycle has paved the way for the development of antivirals for the treatment of EVI [13,14]. Among the previously studied enterovirus inhibitors, the following groups can be distinguished: inhibitors that bind to the viral capsid and prevent its penetration into the cell; capsid binders (pleconaril, pirodavir, vapendavir, pokapavir, disoxaril); inhibitors of viral proteases (rupintrivir and drug AG74/04); viral polymerase inhibitors (ribavirin, gemcitabine, amiloride); and viral ATPase inhibitors (dibucaine, fluoxetine) [15,16,17]. Despite numerous efforts of the scientific community and pharmaceutical companies, no direct-acting antivirals were approved for the treatment of EVI. Therefore, there is an unmet need for new antivirals targeting EVs.

Reactive oxygen species (ROS) generated by cells as a byproduct of oxidative metabolism are able to inactivate DNA, proteins, and lipids, thereby inducing cell death and providing a general defense against many pathogens. Surprisingly, some viruses, including influenza viruses, coronaviruses, herpes viruses, and enteroviruses, use oxidative stress induction via different mechanisms for effective reproduction and enhanced pathogenesis [18,19,20,21]. ROS can be an attractive target for antiviral therapy design [22]. The use of antioxidants (including but not limited to resveratrol, N-acetyl cysteine, quercetin, and their derivatives) for the prevention and treatment of viral diseases has been actively studied [23,24,25]. However, currently, there are no approved drugs inhibiting enterovirus replication through antioxidant mechanisms.

Previously, we described leucoverdazyls as a promising group of substances with antioxidant properties that potently reduce the replication of group B enteroviruses in vitro [26]. Thereafter, we extended the library of leucoverdazyls through directed modifications and investigated virus-inhibiting properties of novel leucoverdazyls against enteroviruses A, B, and C in vitro. According to the results of mechanistic studies, it was suggested that the lead compound targets the 2C protein of coxsackieviruses, a multifunctional non-structural protein with ATPase-dependent helicase and ATPase-independent RNA chaperone activity involved in enteroviral genome replication [13]. The results obtained in this study can be used to develop therapeutic agents to combat EVI.

## 2. Materials and Methods

### 2.1. Compounds

Leucoverdazyls (2-(1-aryl-3-phenyl-5,6-dihydro-4H-1,2,4,5-tetrazin-1-yl)-1,3-benzothiazoles 1–4) were synthesized by alkylation of 1-aryl-5-(1,3-benzothiazol-2-yl)-3-phenylformazans with haloalkanes in alcoholic alkali, followed by cyclization of N-alkyl derivatives as described previously [27]. The structures of leucoverdazyls synthesized are shown in Figure 1 below.

### 2.2. Viruses and Cell Lines

Influenza A virus (IAV, strain A/Puerto Rico/8/1934 (H1N1)), influenza B virus (IBV, strain B/Florida/04/06, Yamagata lineage), Coxsackievirus B3 (CVB3, strain Nancy), Coxsackievirus B4 (CVB4, strain Powers), herpes simplex virus type 1 (HSV1), human adenovirus type 5 (Ad5), and SARS-CoV-2 virus (Wuhan strain) were obtained from the viral collection of the Pasteur Institute (St. Petersburg, Russia). Clinical isolates of ECHO30 (specimen number 4972), Coxsackievirus B5 (GenBank: OQ939946), Coxsackievirus A16 (specimen number 10120), and Coxsackievirus A24 (specimen number 68427) were obtained from The Subnational Polio Laboratory (St. Petersburg, Russia). Virus isolation from stool specimens and identification with enterovirus typing antisera were performed according to the WHO manual [28]. Identification of Coxsackievirus B5 strain was performed by partial sequencing of the VP1 genomic region using EV-specific primers [29]. Influenza A virus (IAV, strain A/Puerto Rico/8/1934 (H1N1)), influenza B virus (IBV, strain B/Florida/04/06, Yamagata lineage), Coxsackievirus B3 (CVB3, strain Nancy), Coxsackievirus B4 (CVB4, strain Powers), herpes simplex virus type 1 (HSV1), human adenovirus type 5 (Ad5), and SARS-CoV-2 virus were obtained from the viral collection of the Pasteur Institute (St. Petersburg, Russia). Clinical isolates of ECHO30 and Coxsackievirus B5 were obtained from the Regional Centre for Epidemiological Surveillance of Poliomyelitis (St. Petersburg, Russia). The following permissive cell lines obtained from ATCC were used in the studies: MDCK (ATCC #CCL-34), Vero (ATCC #CCL-81), RD (ATCC #CCL-136), and A549 (ATCC #CCL-185). Infectious titers (in 50%-tissue culture infection dose, TCID50) were determined in MDCK for IAV; in Vero cells for CVB3, CVB4, CVB5, HSV1, and SARS-CoV-2; in RD cells for ECHO30, CVA16, and CVA24 viruses; and in A549 cells for Ad5. The end-point titration assay was performed using the following procedure. Permissive cells were seeded into 96-well plates in Eagle’s minimal essential medium (MEM) supplemented with 5% fetal bovine serum (FBS, Life Technologies Ltd., Paisley, UK). After 24 h, the media was aspirated, the wells were washed with saline, fresh MEM without FBS was added to the wells, and the cells were infected with ten-fold serial dilutions of viral stocks (100 µL per well, 4 wells for each dilution). Trypsin TPCK (Sigma-Aldrich, St. Louis, MO, USA) was added to the titration media for influenza viruses only. The plates were incubated at +37 °C in 5% CO_2_ and observed daily for cytopathic effect (CPE). Viral titer was calculated in TCID50 using the Spearmen–Carber method.

### 2.3. Antioxidant Activity Testing

Antioxidant activity was evaluated by spectrophotometric monitoring of the hydrogen transfer reaction with a stable chromogen radical, 2,2-diphenyl-1-picrylhydrazyl (DPPH, Sigma-Aldrich, St. Louis, MO, USA), using vitamin C (Vc) as described before [27]. A solution of DPPH in methanol with a concentration of 200 μM was added to a solution of dihydrotetrazines in the same solvent (concentrations 5 to 50 μM). The reaction vessel (test tube) was wrapped in foil and kept for 30 min at 30 °C, and the optical density was measured at λ 517 nm (DPPH absorption maximum). The antioxidant activity (AO) was calculated by the formula
AO = (1 − Atest/Acontr) × 100%,
where Atest is the optical density of a solution containing a compound to be tested and DPPH, and Acontr is the optical density of a solution containing DPPH alone. The half-inhibitory concentration (IC_50_) corresponding to the reduction of the initial DPPH concentration by 50% was determined from the DPPH inhibition percentage plotted against concentrations of compounds 2 through 5 (Figure 1) using the OriginPro 8.5 program (OriginLab Corporation, Northampton, MA, USA) (Model DoseResp).

### 2.4. Cytotoxicity Assay

The microtetrazolium test (MTT assay) was used to study the cytotoxicity of the compounds [30]. Permissive cells were seeded in 96-well plates in Eagle’s minimal essential medium (MEM) supplemented with 10% FBS. After 24 h, the medium was removed, and the wells were washed with saline. Compounds were dissolved in DMSO, and a series of two-fold dilutions of each compound (1000–4 µg/mL) in MEM without FBS were added to the cells in triplicates (200 µL per well). The maximal concentration of DMSO was 0.5%; MEM with 0.5% DMSO was added to cell control wells. Cells were incubated for 24 h or 48 h at 37 °C in 5% CO_2_ and thereafter the MTT assay was performed. The optical density of cells was then measured on a Multiskan multifunctional reader (ThermoFisher Scientific, Singapore) at a wavelength of 540 nm and plotted against the concentration of the compounds to generate the dose–response curve. The 50% cytotoxic dose (CC_50_) of each compound (i.e., the compound concentration that causes the death of 50% of cells in a culture, or decreases the optical density twice as compared to the control wells) was calculated using a four-parameter logistic nonlinear regression model (GraphPad Prism 6). CC_50_ values in μg/mL were then converted into micromoles.

### 2.5. Antiviral Activity Determination

Viral yield reduction assay was used for antiviral activity evaluation. The respective permissive cell lines were seeded in MEM supplemented with 5% FBS in 24-well plates. The next day, the compounds tested were dissolved in DMSO, and a series of three-fold dilutions of each compound (final concentrations 100–1 µg/mL) in MEM without FBS was added to the cells (500 µL per well), followed by incubation (37 °C, 5% CO_2_). After 1 h, the media was discarded, and equal volumes of fresh serial dilutions of each compound (200–2 µg/mL) and viral suspension in MEM without FBS at MOI 0.01 were added to all the wells of the plate (the final volumes were 500 µL per well). In cell control wells, only MEM without FBS was added. In virus control wells, no compounds were added. The plates were incubated at 4 °C for 1 h. Thereafter, the unbound virus was washed away, and again three-fold dilutions of each compound (final concentrations 100–1 µg/mL) in MEM without FBS were added to the wells (1 mL). After 24 h (for enteroviruses and SARS-CoV-2) or 48 h (for HSV1, Ad5, and influenza viruses) of incubation at 37 °C in 5% CO_2_, the infectious titer of viral progeny (in TCID50) for each compound concentration, cell control, and virus control wells were determined in permissive cell lines by end-point dilution assay (described above). 

The titer of viral progeny was plotted against the log concentration of the compounds tested to generate the dose–response curve. The 50% inhibition concentration (IC_50_) of each compound tested (i.e., the compound concentration that decreases the infectious viral progeny titer twice as compared to the control wells) was calculated using a four-parameter logistic nonlinear regression model (GraphPad Prism 6, Boston, MA, USA). IC_50_ values in μg/mL were then converted into micromoles. The selectivity index (SI) was calculated for each compound tested as a ratio of CC_50_ to IC_50_ values. 

### 2.6. Thermostability Assay

The thermostability assay was performed as described previously [31]. The CVB4 strain was chosen because it is sensitive to the pleconaril used in this assay as a reference compound. The compound concentration used in the assay was selected according to the results of preliminary thermostability assays. Briefly, CVB4 Powers (10^4^ TCID_50_) was pre-incubated with the compound (10 µg/mL), pleconaril (10 µg/mL), or an equal volume of MEM (virus control) for 30 min at +37 °C in sterile thin-walled 200 μl PCR-tubes (5 tubes per each treatment condition) in a BioRad CFX PCR-machine (BioRad Laboratories, Inc., Hercules, CA, USA). Then, a thermal gradient of 37–55 °C for 2 min, followed by rapid cooling to 4 °C, was applied. Subsequently, the infectious viral activities of the samples were quantified by end-point dilution assay.

### 2.7. Time-of-Addition Assay

A time-of-addition assay was performed according to the recommendation described earlier; a non-toxic compound concentration not less than 10× its IC_50_ was used [32]. The CVB4 strain was chosen because it is sensitive to pleconaril used as a reference compound. Vero cells were seeded in 24-well plates for 24 h before the beginning of the assay in order to reach 90% confluence. The leader compound was sequentially added in the following time points to the respective individual wells in the plate: (−2), (−1), 0, 2, 4, 6, where (−2) means 1 h before addition of the virus, (−1)—addition of the virus, 0—completion of viral sorption on the cell surface, and 2, 4, 6—in 2, 4, or 6 h after virus sorption. At timepoint (−1), a suspension of 10^6^ TCID50 of CVB4 was added to all the wells (except cell control), and the plate was incubated at +4 °C for 1 h in order to synchronize the infection in all conditions. Afterward, the unbound virus was washed off, and the plate was returned to +37 °C. The capsid binder pleconaril was used as a reference compound. Eight hours after the completion of virus sorption, the experiment was stopped, and the infectious viral titer was measured in each well using end-point titration in Vero cells.

### 2.8. Selection and Analysis of the Drug-Resistant Strain

In order to study the development of resistance to the lead compound, the CVB3 virus (Nancy) was serially passaged in Vero cells in the presence of increased concentrations of the compound. CVB3 Nancy was selected as a model virus because its nucleotide sequence is annotated and available from GenBank. Cells were infected with the virus and incubated for 2–3 days (37 °C, 5% CO_2_) until a cytopathic effect was observed. The culture supernatants were used for sequential selection. In total, nine passages were performed using the following **1a** concentrations: 0.5, 1, 2, 3, 4, 5, and 6 µg/mL, followed by two passages at 7 µg/mL to obtain the resistant (R) variant. The wild-type (WT) virus was passaged in Vero cells in the absence of **1a**. The values of IC_50_ for original, WT, and R viruses were further determined by viral yield reduction assay. 

Three viral variants (original, WT, R) were plaque purified, and full genomes of three clones from each virus were sequenced. Viral RNA was extracted using the Ribo-prep kit (Amplisense, Moscow, Russia). After reverse transcription using an MMLV RT kit (Evrogen, Moscow, Russia) and amplification of cDNA using the high fidelity polymerase Tersus plus (Evrogen, Moscow, Russia) and CVB3-specific primers, purified PCR-products were analyzed on ABI-3500 XL Genetic Analyser (Thermo Fisher Scientific^®^, Cambridge, UK) using BigDye^®^ Terminator v3.1 Chemistry and POP-7™ polymer (Thermo Fisher Scientific^®^, UK). CVB3-specific primers used for cDNA amplification and sequencing are listed in Appendix A (primer sequences were adapted from the publication by Liu et al.) [33]. Chromatograms were converted into contigs using Unipro Ugene free software (version 45.1, Novosibirsk, Russia) [34]. The sequences were aligned to a reference CVB3 sequence (GenBank: M16572.1). The nucleotide sequences were translated into amino acids by free online software (https://web.expasy.org/translate/) (accessed on 12 February 2022) [35]. 

### 2.9. Growth Kinetics

The in vitro growth kinetics of CVB3-WT and -R variants were determined in Vero cells (6-well tissue culture plates, 1 × 10^6^ cells/well). At the zero timepoint, cells were infected with the respective viral variants (MOI 0.001). After 2 h of virus absorption, cells were washed three times with MEM to remove non-adsorbed virus, and 3000 μL of medium either with or without **1a** was added to corresponding wells. A concentration of 1a (7 ug/mL) was selected as the highest concentration used in the resistance selection study. At 8, 24, 36, 48, 60, and 72 h post-infection, an aliquot of culture supernatants from each well was collected to quantify the number of infectious viral particles at each time point by end-point titration in 96-well tissue culture plates.

### 2.10. Transmission Electron Microscopy (TEM)

Vero cells in 6-well plates were incubated with CVB3 virus, MOI = 100, at 4 °C for 1 h. Unbound virions were removed by washing the cells twice with cold MEM; a medium containing 100 µM of **1a** was added. This high concentration of **1a** was selected based on the assumption that it should completely prevent CPE development upon cell infection with 100 MOI CVB3. Cells were incubated for 3 h at 36 °C in 5% CO_2_. In wells with control virus, MEM without **1a** was added. After incubation, cells were collected from the wells, transferred into tubes, and centrifuged at 2000× *g* for 15 min. Cell pellets were fixed with 1.5% glutaraldehyde in PBS overnight, followed by post-fixation with 1.5% OsO4 for 1 h and uranyl acetate for 45 min at room temperature. They were then dehydrated in graded acetone and embedded in Epon/Araldit resin (Serva Feinbiochemica, Heidelberg, Germany). Thin sections (90 nm) were stained with lead citrate and examined in a JEM-100S electron microscope (JEOL, Tokyo, Japan).

### 2.11. Computer Modeling 

The model of the 2C protein was prepared using AlphaFold v.2.3 software. Molecular docking for modeling the interaction between **1a** and 2C was done by Hex online server (http://hexserver.loria.fr/, accessed on 1 May 2023) [36].

### 2.12. Statistics

All in vitro experiments were repeated three times. The results are represented as mean ± standard deviation (SD). Viral titers were plotted against the logarithm of concentration, and the IC_50_ values for each virus were calculated using GraphPad Prism (v.8.0) software using four-parameter logistic curves (4PL). The results of TEM were analyzed using χ^2^ method (Statistica 8.0 software).

## 3. Results

### 3.1. Leucoverdazyls Are Potent Inhibitors of CVB3 Nancy at Low Micromolar Concentrations

At the beginning of the study, the virus-inhibition activity of leucoverdazyls in viral yield reduction assay was determined. The cytotoxicity of the compounds for permissive cell lines used was evaluated by MTT assay. The results are summarized in Table 1 below. Pleconaril was used for comparison as a reference compound due to the fact that CVB3 Nancy is a pleconaril-resistant strain [37].

As can be seen from the data presented in the table, the least toxic compounds were the following: **1b**, **2b**, **3b**, **3d**, and **4d**. Two of them (**1b**, **2b**) contain a methoxy group in the aromatic fragment. Halogen-containing compounds **2d** and **2e** showed the greatest toxicity. In the viral yield reduction assay, a large majority (57%) of leucoverdazyls from the library (11 out of 19 tested) demonstrated remarkable anti-enteroviral activity in vitro against pleconaril-resistant CVB3 strain in comparison to pleconaril. SI values for these compounds exceeded 10, which is indicative of high anti-enteroviral potential. Compounds **1a**–**1c** exhibited the most pronounced antiviral activity, with IC_50_ values much lower than that of pleconaril. It should be noted that the structure of these compounds lacks a substituent at position 6 of the tetrazine ring. In addition, the best values of the selectivity index were also noticed for compounds that do not contain a substituent in this position of the tetrazine ring (compounds **1a**–**1d**), as well as for compounds containing the least bulky substituent (Me)—compounds **2a, 2b**. Compound **1a** with the lowest IC_50_ value (2.7 µM) and the highest SI (230) was selected as the leader for further study. 

In order to trace the effect of the substituent at position 6 of the tetrazine ring on the antioxidant activity of dihydrotetrazines, a DPPH test was performed in the series of **1a,b**, **2a,b**, **3a,b**, and **4a,b**. The results are presented in Figure 2.

As can be seen from the data obtained, the most pronounced antioxidant activity was detected for compounds **1a** and **1b**, and their antioxidant potentials were even superior to vitamin C used as the reference compound. Therefore, the data show that the addition of a substituent in the sixth position of the tetrazine ring has a negative impact on the antioxidant activity of dihydrotetrazines in vitro. 

The data on antioxidant activity indicate that compounds without substituent in the sixth position of the tetrazine ring, as well as compounds with a methoxy group in the aromatic fragment at N1, turned out to possess the highest antioxidant activity.

### 3.2. ***1a*** Possesses Wide-Range Activity against Group A, B, and C Enteroviruses

To assess the prospects for further development of the most potent compound (**1a**), the spectrum of its anti-enteroviral activity was assessed against a panel of group A, B, and C enteroviruses including both Coxsakievirus B4 (strain Powers) and patient isolates in viral yield reduction assay. The following patient viral isolates were used: CVA16, CVB5, ECHO30, and CVA24. Among them, CVA16 is a common agent of HFMD, while CVB5 and ECHO30 have also been reported to be associated with HFMD [38,39,40]. The results are presented in Table 2 below. Guanidine hydrochloride targeting the initiation step of viral RNA synthesis was used as a reference drug [41].

Compound **1a** showed significantly higher antiviral activity towards other strains of group A, B, and C enteroviruses in comparison to the reference compound guanidine hydrochloride, though guanidine hydrochloride was less toxic. We further investigated whether **1a** is capable of inhibiting the life cycle of other phylogenetically distinct RNA or DNA viruses. Compound **1a** was tested against influenza (ssRNA-negative enveloped virus), HSV1 (dsDNA enveloped virus), Ad5 (dsDNA non-enveloped virus), and SARS-CoV-2 (ssRNA-positive enveloped virus) in viral yield reduction assay (Table 3).

According to the results, compound **1a** showed only modest activity against influenza viruses, Ad5, and HSV1. Surprisingly, however, it inhibited replication of another ssRNA-positive enveloped virus, namely SARS-CoV-2. This leads us to the assumption that **1a** targets some biological entity (protein or process) common and important for both enterovirus and coronavirus life cycles.

### 3.3. ***1a*** Does Not Increase Virion Thermostability and Inhibits Late Stages of the CVB4 Life Cycle

The plausible mechanism of action for **1a** was studied using in vitro assays. We addressed whether **1a** has capsid-binding properties using a thermal stability assay using CVB4 (strain Powers). It is known that capsid binders (pleconaril and its derivatives) directly interact with the capsid of enteroviruses and stabilize its structure, thereby preventing the virus from entering the host cell. This interaction increases the resistance of the viral capsid to a short-term temperature increase, and the heated virus retains its ability to infect a permissive cell line. The results are presented in Figure 3 below.

In the virus control, as well as when test drug **1a** was mixed with the virus, no infectious particles were detected when heated above 45 degrees. As expected, the reference capsid-binding drug pleconaril had a thermostabilizing effect on the Coxsackie B4 virus, ensuring the presence of infectious particles even when heated to 55 degrees (the highest temperature used). Thus, it was concluded that **1a** does not belong to the capsid-binding group of inhibitors.

Next, we focused on the stage in the viral cycle when **1a** demonstrated the highest inhibitory activity in the time-of-addition assay. CVB4 was propagated in Vero cells with the addition and removal of **1a** at distinct time points before or after the zero point when the virus was added. After one cycle of replication (8 hpi), the infectious titer of viral progeny was determined in the end-point dilution assay. The titer of viral progeny versus the interval of **1a** presence in the media is presented in Figure 4.

The most pronounced inhibitory effect of **1a** was demonstrated if it was present in the culture medium starting from −2 to 4 h post-infection (hpi). Inhibition of viral replication was not observed if the substance was added later than 6 h after the adsorption of the virus. Pleconaril used as a reference compound demonstrated the highest activity between −2 and 0 h, as expected for early-stage inhibitors. The results obtained suggest that the substance acts on steps involved in viral replication.

The Coxsackievirus life cycle is relatively short (6–8 h) and has been extensively studied previously [13,14]. Briefly, after receptor-mediated endocytosis, viral genomic RNA is translated into a polyprotein, which in turn is proteolytically processed by viral 2Apro and 3Cpro to release viral proteins, and viral RNA replication begins. Viral RNA replication is performed via a dsRNA intermediate in specialized replication organelles. Nascent viral (+)RNA is encapsidated by structural proteins to form new virions, which are released either lytically or non-lytically (in autophagic vesicles). 

According to previously obtained results, viral RNA replication is initiated 2–3 h after infection, and translation of viral capsid proteins is detectable as early as after 4 hpi [42]. Therefore, the inhibitory effect of **1a** spans the following stages of the viral life cycle: cell attachment, penetration, genomic RNA transcription, proteolytic processing, and RNA replication.

### 3.4. Transmission Electron Microscopy

In order to visualize the effect of **1a** on viral morphogenesis, we performed an analysis of the ultrastructure of CVB3-infected cells in the presence of **1a** versus non-infected cells and infected cells without treatment (Figure 5A–C). A total of 49 and 61 microphotographs of CVB3-infected cells treated with **1a** and infected cells without treatment, respectively, were analyzed. 

In infected cells without treatment, in 38 out of 61 cells observed, typical cytoplasmic membranous vesicles were detected, representing replication organelles typical for enterovirus infection [43]. Treatment of cells with **1a** abrogated these changes in the cytoplasm. As can be seen, compound **1a** eliminates signs of viral replication in infected cells (no replication organelles were visible in 47 out of 49 examined cells). Therefore, **1a** affects the formation of replicative organelles in CVB3 infected cells, df = 1, χ^2^ = 39.79, *p* < 0.05 (Figure 5D).

### 3.5. ***1a***-Resistant Strain Selection and Its Genomic and Phenotypic Characteristics

In order to assess the genetic barrier to resistance development to **1a**, we further passaged CVB3 (Nancy strain) in Vero cells at increasing concentrations of **1a**, evaluated the emerging resistance level, and identified amino acid substitutions. Three viral strains were analyzed: “original” CVB3 from the bank, which was used to generate a “wild-type” virus (CVB3 WT, passaged without **1a** in Vero cells), and “resistant” strain (CVB3 R, passaged at increasing concentrations of **1a**). After nine subsequent passages of the virus in cell culture, the IC_50_ of **1a** was determined to be 12.9 μM for the resistant strain, which was 7-fold higher than that of the original virus (IC_50_ = 1.8 µM), and higher than wild-type virus (IC_50_ = 0.48 µM). Therefore, **1a** stimulates the selection of resistance to the CVB3 virus, suggesting its direct antiviral activity and a virus-specific target (Figure 6). We also investigated the growth characteristics of the resistant virus in comparison to the wild-type one in vitro (Figure 7). In the presence of **1a**, the resistant strain was able to effectively propagate in contrast to the wild-type virus. Nevertheless, without **1a**, the growth speed of the resistant strain was significantly lower than that of the wild-type virus during the first 48 h (*p* < 0.05 by the Mann–Whitney U-test). 

After the resistant viral variant was obtained, viruses were plaque purified, and full genomes of three clones from each viral type (initial, wild-type, **1a**-resistant) were sequenced. Their nucleotide sequences were translated to localize amino acid substitutions. After a comparison of the amino acid sequences, substitution S1209I was identified. Since position 1209 of viral polyprotein corresponds to the 2C protein, here and further we use amino acid numeration corresponding to this specific protein (‘position 109’, instead of 1209). AlphaFold software was used to generate the complete structural model of 2C protein. Based on the model, the position 109 in 2C was mapped (Figure 8).

Therefore, with sequential passage of the Coxsackievirus in the presence of **1a**, a decrease in the sensitivity of the virus to **1a** occurs. This is accompanied by a deterioration in the growth characteristics of the resistant virus and the appearance of mutation in the 2C protein. The 2C protein is a multifunctional enteroviral protein, which among other processes participates in viral genome replication [13]. The role of 2C protein in the enteroviral life cycle is described in detail in the discussion section below. This suggests a possible influence of **1a** on processes associated with replication of the viral genome.

### 3.6. Molecular Modeling

Further, we compared the localization of the S109I amino acid substitution in the 2C protein of **1a**-resistant virus with the localization of the probable binding site of **1a**. As shown in Figure 9, the binding site for **1a** appeared to be located in close proximity to the S109I substitution, thus corresponding to the target for **1a** among virus-specific proteins.

## 4. Discussion

Enteroviruses represent a clinically important group of human pathogens with neither vaccine nor direct antivirals available for enteroviral infection management. In the present study, we showed the high antiviral potential of a novel class of compounds, leucoverdazyls, against enteroviruses A, B, and C, including both laboratory strains and patient isolates, and described its plausible mechanism of action. We also assessed the possibility of selection of a viral variant resistant to lead compound **1a** and characterized its properties including fitness, susceptibility to the inhibitor, and genomic composition. Our results suggest that the compounds of this class exert their virus-inhibiting activity at the early stages of the viral cycle (before 4 hpi). The drug-resistant Coxsackie B3 viral variant featured an IC_50_ value seven-fold higher than that of the wild-type virus. Amino acid substitution S109I in the 2C viral protein was detected in the resistant virus. Molecular docking of **1a** to the 2C protein showed that the ligand and substituted amino acid are localized at the same domain of 2C.

The life cycle of enteroviruses has been described. Following cell entry, capsid disassembly, and exposure of viral RNA to the cellular translational system, the viral genome is translated into a single polyprotein which is further processed by viral protease into structural and non-structural virus-specific proteins [44]. Double-stranded RNA is an essential intermediate in the process of viral genome replication in cellular cytoplasm wherein host-pathogen recognition receptors (PRRs), in particular dsRNA sensors, provide protection from invading viral pathogens [45]. dsRNA is one of the most important pathogen-associated molecular patterns (PAMPs). Therefore, in order to avoid contact with PRRs and further progression of innate antiviral immune reactions, enteroviruses induce the formation of single- and double-membrane vesicles called replication organelles (ROs). Therein, processes of RNA replication and virion assembly take place being protected from host cell antiviral defense mechanisms [46,47].

To provide large amounts of membrane required for RO formation, enteroviruses developed numerous ways to manipulate host cell pathways for biogenesis and functionality of the membranous structures in infected cells [43,48]. Some viral proteins must, therefore, be associated with membranes to properly execute their function and realize the viral life cycle. The 2C protein is one of the most conserved proteins within the Picornaviridae family with multiple functions [49,50]. This non-structural protein of 322–330 amino acids is involved in virus uncoating, host cell membrane binding, and rearrangement, formation of the viral cytoplasmic replication vesicles, RNA binding and RNA synthesis, and possibly encapsidation [51,52,53]. However, in infected cells, it is localized in Golgi-related membranes [44]. It possesses ATP-dependent RNA helicase and ATP-independent chaperoning activities [52]. 

Recently, the 2C protein was demonstrated to possess ATPase-independent nuclease activity with a preference for polyU ssRNAs [54]. The amino acids essential for RNAse activity have been mapped to the central pore of the hexameric ring. In addition, enterovirus 2C proteins affect the NF-κB signaling pathway, one of the most important mechanisms of innate antiviral immunity, by recruitment of protein phosphatase (PP1) for suppression of IKKβ phosphorylation [55] and by direct binding to IKKb, as well as with p65 and MDA5 [56,57]. In addition, EV71 2C induces the degradation of another innate immunity protein, APOBEC3G, by its ubiquitination utilizing the autophagy-lysosome pathway [58]. 

For proper action, 2C has to be oligomerized into a ring-shaped hexamer [59,60,61]. However, detailed structural and functional characterization of 2C is impeded by the presence of an N-terminal amphipathic helix which makes the protein insoluble and impossible to crystallize [62,63,64]. Most of the structural studies, therefore, have been done with a soluble fragment of 2C covering the ATPase domain, a cysteine-rich zinc finger, and a C-terminal helical domain. In our study, we took advantage of the predictive ability of AlphaFold software to build a complete model of CVB3 2C protein based on its amino acid sequence. 

The **1a** resistance-associated mutation S109I was shown to be located within the interface between the head of 2C and its tail containing amphilin, i.e., the possible domain where 2C is inserted in cytoplasmic membranes. To the best of our knowledge, no 2C mutations that confer viral resistance to clinical or experimental compounds were detected in the domain we described in our study. No specific function of 2C, therefore, can be hypothesized to be altered by **1a**. In 2000, Klein et al. demonstrated that due to 2C mutations in or near the NTP binding domain, resistance to 2-(α-hydroxybenzyl)-benzimidazole (HBB) and guanidine was achieved [65]. Viruses with lowered sensitivity to other benzimidazole derivatives, TBZE-029 and MRL-1237, guanidine-HCl, and hydantoine, were obtained bearing mutations in positions 64, 65, 120, 125, 133, 142, 143, and several others located at an even longer distance from the position 109 we found [66,67]. Based on the docking results, none of these mutations could alter the binding of the leucoverdazyl derivative to 2C (Figure 9). The mechanism of action of **1a** is, therefore, distinct from that of all described compounds including guanidine-HCl and benzimidazole derivatives.

As indicated above, the N-terminus of 2C (amino acids 1–125) interacts with all isoforms of the PP1 catalytic subunit through a PP1-docking motif [68]. One possibility is, therefore, that **1a** interferes with this binding thus preventing virus-induced blockade of the NF-κB pathway. On the other hand, binding of the N-terminus of EV71 2C with host protein reticulon 3 (RTN3) was shown to be necessary for the synthesis of viral proteins and replicative double-stranded RNA [69]. The protein site responsible for this interaction was mapped within amino acids 10–27, with isoleucine 25 having the highest importance for binding. This part of 2C, however, is located rather far from the mutated amino acid 109 and **1a** binding site. Therefore, alteration of 2C-RTN3 interaction by **1a** is of low probability, or indirect. One more possibility is that, since the N-terminal amphipathic helix of 2C binds to cellular lipid droplets to build RO membranes [70], **1a** could potentially interfere with this interaction too. 

Importantly, data on antioxidant activity indicate that compounds without substituent in the sixth position of the tetrazine ring, as well as compounds with a methoxy group in the aromatic fragment at N1, turned out to be the most effective antioxidants. Interestingly, while possessing the highest antioxidant activity (Figure 2), compounds **1a** and **1b** also demonstrated the best virus-inhibiting properties toward CVB3 in vitro (Table 1). As shown previously, enteroviruses can use ROS-based signaling and metabolic processes for their efficient propagation within the cell [18,19,20,21]. It cannot be ruled out, therefore, that **1a** has a dual mechanism of virus-inhibiting activity, potentially being a multitarget compound. Few compounds, however, have been studied in this regard, and this issue should be addressed in further experiments.

## 5. Conclusions

In conclusion, we have identified a novel class of anti-enteroviral compounds, leucoverdazyls, differing in structure and mechanism of action from all previously described viral inhibitors. Further structural and functional studies are necessary to fully understand the mechanism of antiviral activity of leucoverdazyls and the molecular basis of resistance formation.

## Figures and Tables

**Figure 1 pathogens-13-00410-f001:**
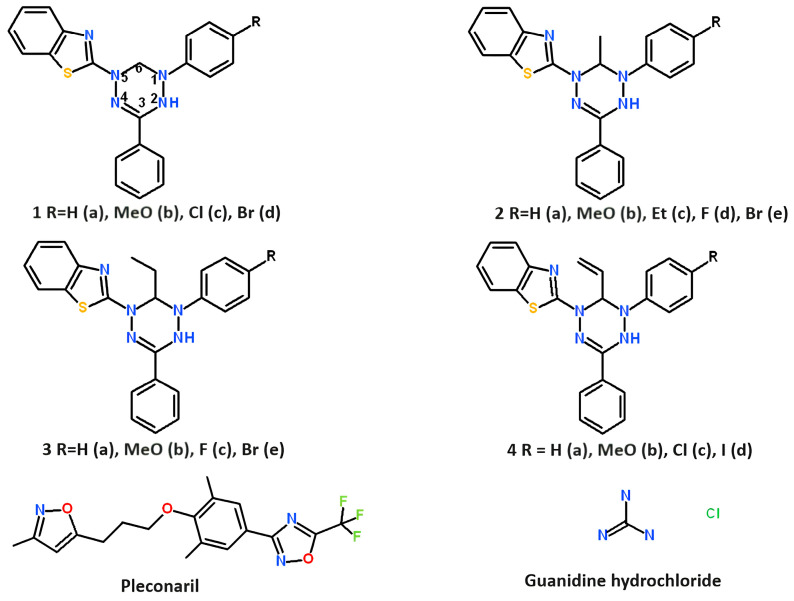
Structures of leucoverdazyls tested in the study and reference compounds: pleconaril and guanidine hydrochloride. Pleconaril was kindly provided by Dr. V. A. Makarov (Research Center of Biotechnology RAS, 33-1 Leninsky Prospect, 119071, Moscow, Russia). Guanidine hydrochloride was bought from Dia-M Ltd. (Moscow, Russia).

**Figure 2 pathogens-13-00410-f002:**
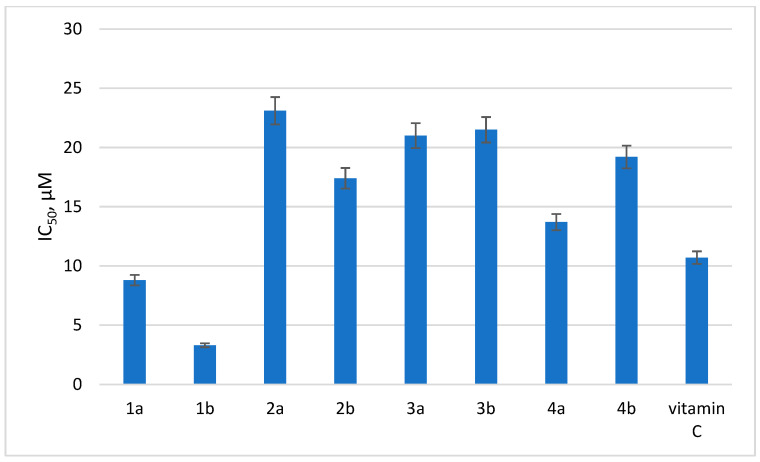
Antioxidant activity of selected dihydrotetrazines. Presented are IC_50_ values (µM) for the antioxidant activity of the tested compounds by DPPH assay. Vitamin C was used as a reference.

**Figure 3 pathogens-13-00410-f003:**
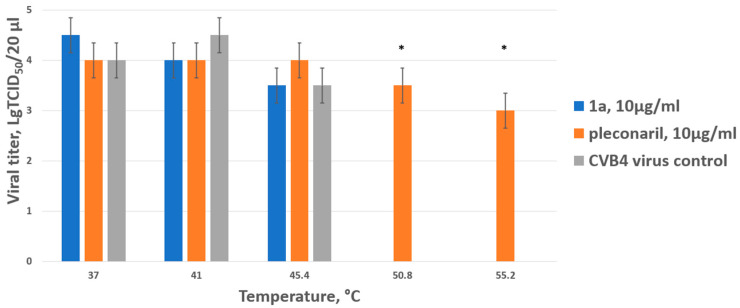
Thermostabilizing properties of **1a** in comparison to those of pleconaril. Values are the mean ± SD of three independent experiments. The legend shows the concentration of each compound tested. The asterisk indicates the significance of the difference in viral titer for pleconaril at 51 °C and 55.2 °C relative to the virus control, *p* < 0.05 by Mann–Whitney U-test.

**Figure 4 pathogens-13-00410-f004:**
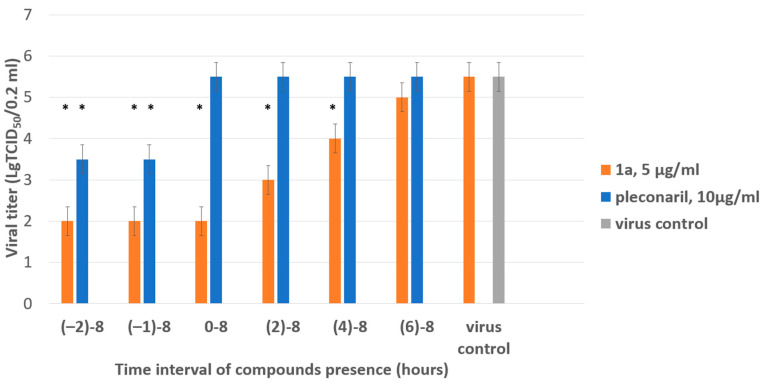
Results of time-of-addition assay for **1a**. The activity of compound **1a** against the Coxsackie B4 virus (Powers strain) depending on the time of addition to a permissive cell line upon CVB4 infection. Vero cells were infected with CVB4 (−1 h), and **1a** (5 μg/mL) was added at the indicated time points (in hours) either before the virus (−2 h), concomitantly with the virus (−1 h), or after (0, 2, 4, 6 h) infection, where 0 corresponds to the moment of completed virus absorption on the cell surface. The infectious activity of the viral progeny was evaluated by end-point titration in the Vero cells in lg TCID_50_/0.2 mL. Pleconaril (10 μg/mL) was used as a reference compound. Values are presented as the mean ± SD of three independent experiments. An asterisk indicates a significant difference in viral titer for **1a** and pleconaril relative to the virus control, *p* < 0.05 by the Mann–Whitney U-test.

**Figure 5 pathogens-13-00410-f005:**
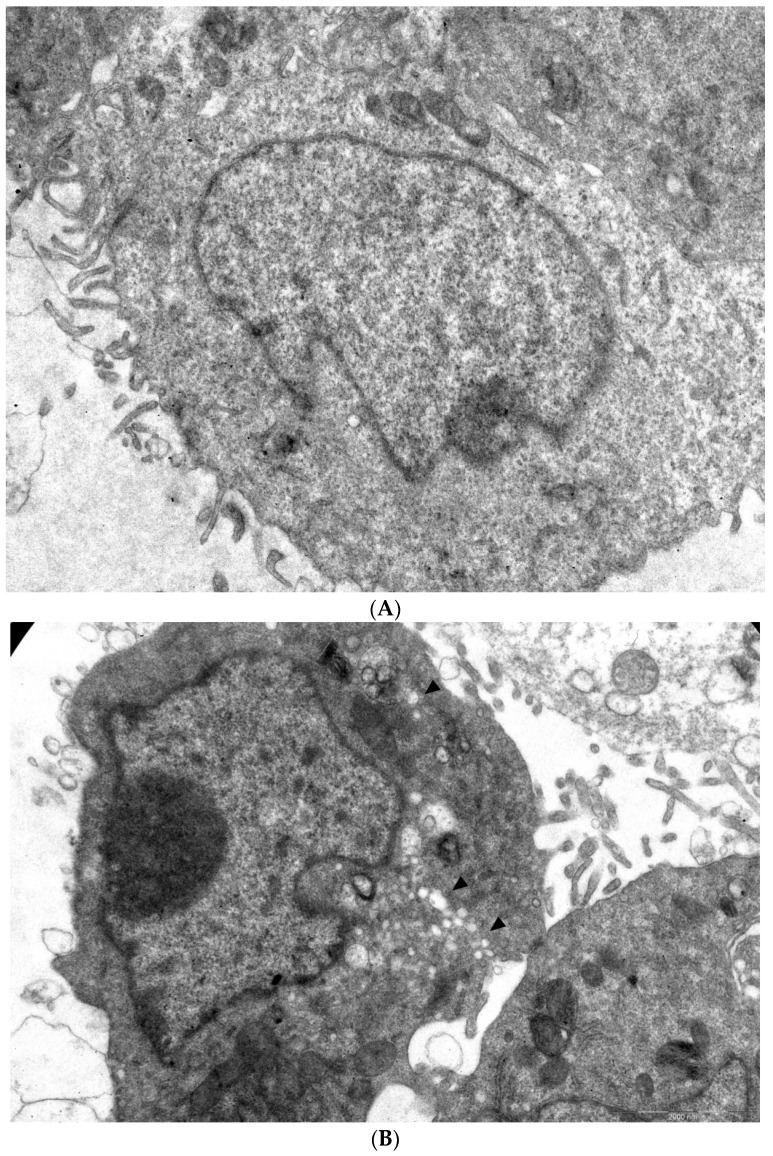
Ultrastructure of Vero cells infected by CVB3 revealed by transmissive electron microscopy, representative microphotographs. (**A**) Intact cell. No vacuoles or replication organelles are visible within the cytoplasm. (**B**) CVB3-infected cell. Numerous vacuoles representing virus-specific replication organelles are indicated by arrowheads. (**C**) CVB3-infected cell in the presence of 100 μM compound **1a**. No morphological signs of viral replication can be seen. (**D**) Statistical analysis of cell numbers with and without signs of viral replication in **1a** treated CVB3 infected group versus CVB3 infected non-treated group, df = 1, N = 110, χ^2^ = 39.79, *p* < 0.05.

**Figure 6 pathogens-13-00410-f006:**
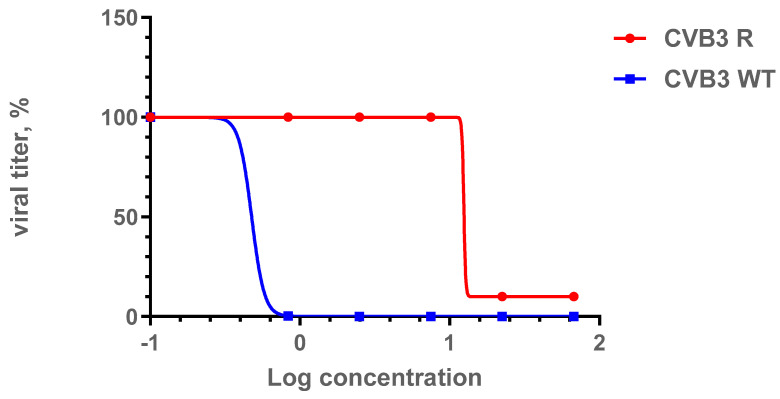
Comparison of IC_50_ values for the CVB3 R and CVB3 WT strains. Presented are the results of the viral yield reduction assay for two CVB3 strains: wild-type and resistant virus propagated in the presence of **1a**. The 4PL were fitted using GraphPad Prism 6. Viral titer is represented in % relative to virus control.

**Figure 7 pathogens-13-00410-f007:**
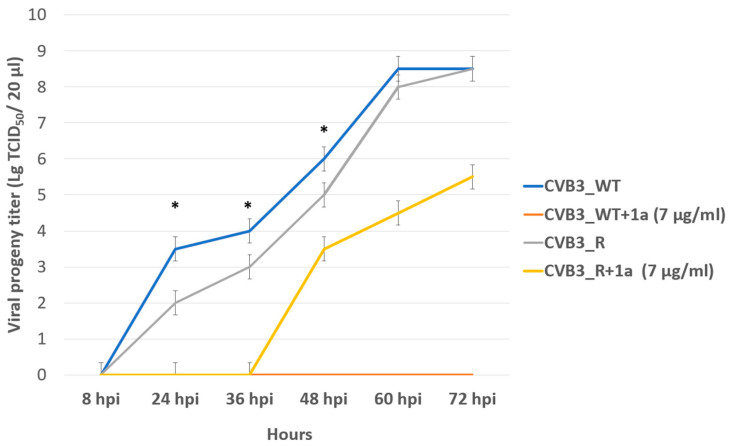
Propagation kinetics of CVB3 WT and CVB3 R strains in Vero cells with and without **1a**. Viral progeny titer is plotted versus incubation time. Multistep growth curves are presented. An asterisk indicates a significant difference in virus titer, *p* < 0.05 by the Mann–Whitney U-test.

**Figure 8 pathogens-13-00410-f008:**
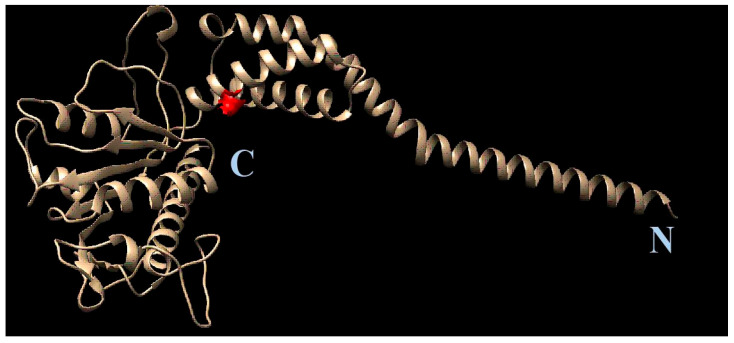
Position of S109I mutation in 2C protein. The amino acid position is depicted in red. The C- and N- termini of the protein are marked with C and N, respectively.

**Figure 9 pathogens-13-00410-f009:**
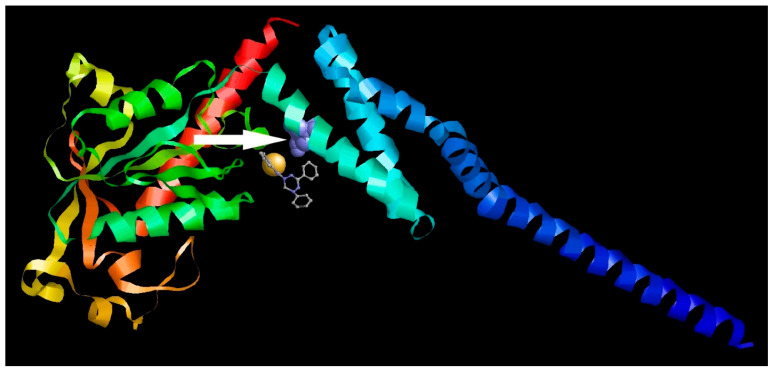
Colocalization of the **1a** binding site and the S109I amino acid substitution in the 2C protein of **1a**-resistant Coxsackievirus B3. Different protein chains are marked with different colors. The substitution is indicated by the arrow.

**Table 1 pathogens-13-00410-t001:** Cytotoxic and virus-inhibiting properties of leucoverdazyls against Coxsackie B3 virus in vitro.

Compound Number	CC_50_, µM ^a^	IC_50_ µM ^b^	SI ^c^
**1a**	619.9 ± 52.7	1.8 ± 0.3	>230
**1b**	>1347	5.4 ± 1.1	>250
**1c**	320.9 ± 28.2	7.4 ± 2.2	43
**1d**	886.4 ± 90.4	6.2 ± 1.9	142
**2a**	>324.7	6.49 ± 0.8	>50
**2b**	2048.2 ± 198.5	24.1 ± 3.2	85
**2c**	>302.7	>121.1	>2
**2d**	49.6 ± 3.7	>49.6	<1
**2e**	69.1 ± 5.9	>69.1	<1
**2f**	>301.2	12.1 ± 1.9	>25
**3a**	>313.3	>125.3	>2
**3b**	2331.1 ± 25.3	60.1 ± 7.2	>38
**3c**	359.7 ± 41.7	71.9 ± 6.4	5
**3d**	2309.5 ± 19.7	>231.0	>10
**3e**	945.5 ± 89.6	>209.6	>4
**4a**	>314.8	>125.9	>2
**4b**	>292.7	>117.1	>2
**4c**	>290.7	27.9 ± 3.3	10
**4d**	1529.6 ± 17.2	51.6 ± 6.4	29
**Pleconaril**	656.3 ± 58.4	15.2 ± 1.6	43

^a^ CC_50_ is the cytotoxic concentration, the concentration resulting in the death of 50% of the cells (CC_50_ were evaluated after 24 h of Vero cells incubation with compound only); ^b^ IC_50_ is the 50% virus-inhibiting concentration, the concentration leading to 50% inhibition of virus replication; ^c^ SI is the selectivity index, the ratio of CC_50_/IC_50_. The values for CC_50_ and IC_50_ are presented as the mean ± SD of three independent experiments.

**Table 2 pathogens-13-00410-t002:** Activity of **1a** against a panel of group A, B, and C enteroviruses.

Virus	Cell Line Used	CC_50_, µM ^a^	IC_50_ µM ^b^	SI ^c^
	Compound **1a**			
CVA16	RD	673.3 ± 60.1	0.8 ± 0.2	841
CVB5	Vero	619.9 ± 52.7	1.5 ± 0.3	413
ECHO30	RD	673.3 ± 60.1	1.6 ± 0.4	420
CVA24	RD	673.3 ± 60.1	0.9 ± 0.3	747
CVB 4 (strain Powers)	Vero	619.9 ± 52.7	1.7 ± 0.5	364
	Guanidine hydrochloride			
CVA16	RD	>10,471.2	336.8 ± 28.3	>31
CVB5	Vero	>5235.6	473.7 ± 33.4	>10
ECHO30	RD	>10,471.2	125.6 ± 18.2	>83
CVA24	RD	>10,471.2	314.2 ± 20.1	>33
CVB4 (strain Powers)	Vero	>5235.6	495.7 ± 26.5	>10

^a^ CC_50_ is the cytotoxic concentration, the concentration resulting in the death of 50% of the cells (CC_50_ were evaluated after 24 h of incubation of RD or Vero cell lines with compound only); ^b^ IC_50_ is the 50% virus-inhibiting concentration, the concentration leading to 50% inhibition of virus replication; ^c^ SI is the selectivity index, the ratio of CC_50_/IC_50_. The values for CC_50_ and IC_50_ are presented as the mean ± SD of three independent experiments.

**Table 3 pathogens-13-00410-t003:** Activity spectra of **1a** against RNA and DNA viruses of various structures.

Virus	Cell Line Used	CC_50_, µM ^a^	IC_50_ µM ^b^	SI ^c^
Influenza A/Puerto Rico/8/34	MDCK	850.2 ± 78.3	38.7 ± 2.9	22
Influenza B/Florida/04/0 6	MDCK	850.2 ± 78.1	40.5 ± 5.1	21
HSV1	Vero	450.7 ± 51.2	46.3 ± 4.4	10
Ad5	A549	500.6 ± 45.7	43.1 ± 3.8	12
SARS-CoV-2	Vero	656.3 ± 70.4	4.5 ± 1.2	146

^a^ CC_50_ is the cytotoxic concentration, the concentration resulting in the death of 50% of the cells (CC_50_ were evaluated after 48 h or 24 h of incubation with compound only depending on the life cycle length of the particular virus tested); ^b^ IC_50_ is the 50% virus-inhibiting concentration, the concentration leading to 50% inhibition of virus replication; ^c^ SI is the selectivity index, the ratio of CC_50_/IC_50_. The values for CC_50_ and IC_50_ are presented as the mean ± SD of three independent experiments.

## Data Availability

All data will be made available by the authors on request.

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
