# Peer review of "Leucoverdazyls as Novel Potent Inhibitors of Enterovirus Replication"

_pathogens, 2024, doi:10.3390/pathogens13050410_

Round 1

Reviewer 1 Report

Comments and Suggestions for Authors

Summary

This paper titled “Leucoverdazyls as novel potent inhibitors of enterovirus replication” aims to demonstrate the capacity of leucoverdazyls to act as a novel class of antiviral compounds against enteroviruses. They selected a lead compound (1a) after screening a group of 19 compounds and this lead compound was tested mainly against CVB3 and CVB4 using a variety of assays to prove its antiviral potential. They suggest a plausible mechanism of antiviral activity for 1a via binding to viral protein 2Cpro thereby interfering with viral genome replication.

The manuscript is written with proper flow and sufficient clarity. The discoveries are relevant and novel for the field. There are fewer clarifications/explanations that the authors need to include for improving the clarity as well as the quality of the manuscript (suggested below). The methods section has sufficient details for reproducing the results. The manuscript is scientifically sound, and the experimental design is appropriate to test the hypothesis. The conclusions are consistent with the evidence and arguments presented. It is better if the quality of the images is improved and if the graphs are not directly used from excel.

Major revisions:

Results

The two viruses CVB3 and CVB4 have been used in different types of assays. In the beginning CVB3 was used to test the list of compounds (section 3.1). Later, CVB4 was tested against compound 1a (section 3.2) selected out of the list. For thermostability and viral inhibition assays (section 3.3), CVB4 has been used while for TEM images (section 3.4), CVB3 has been used. The authors have been using these 2 viruses interchangeably among experiments. However, they haven’t reasoned why they don’t use one of the viruses throughout the experiments. It is appropriate if authors explained within text why they had to use that specific virus out of the 2 for that specific experiment. In addition, I observed that the CC50 values indicated for CVB3 and CVB4 are exactly the same while the IC50 values are different (1.8±0.3 and 1.7±0.5, respectively). It needs to be double checked to see if the similar CC50 values are indicated by mistake.

The concentration of compound 1a used in different assays varies among each other. The authors should provide a logical explanation why they used these specific concentrations instead of using one specific concentration. Eg: 5 ug/ml (Figure 4), 10 ug/ml (Figure 3), 7 ug/ml (Figure 7). I suggest generating high quality images using a software like adobe illustrator instead of using the
graphs generated in excel directly.

Line 442: I doubt if the authors are capable of arriving at the conclusion that “1a has an influence on processes associated with replication of the viral genome”. They know that 2C gets a point mutation and possibly this mutation affects the binding of 1a to 2C thereby reducing the inhibitory effect of 1a. When the authors comment on the viral genome replication here, they should at least briefly state what is the role of 2C protein in viral genome replication, even though they explain in greater detail within the discussion section later.

If the authors need to confirm that 1a exerts its effects via binding to 2C, I suggest they can transfect the cells infected with 1a-resistant CVB3, a plasmid expressing the wt 2C protein to see if the resistance of CVB3 towards 1a is reversed.

Discussion

Line 457: It is more appropriate if stated as “plausible mechanism of action” instead of “mechanism of action” because the binding of drug at close proximity to the mutated site is only proved by MD simulations. There is a possibility that 2C is not the only target of 1a. Also, the mutation of 2C can have an indirect effect on rendering the resistance instead of a direct effect from 2C-1a interaction. However, given that the binding of 2C and 1a has not been proven using structural or biophysical studies, the proposed mechanism of action should be stated as something plausible.

Minor revisions:

Introduction

Lines 49-50: Better to use standard terms of classification for enteroviruses. Refer EV-A and EV-B as species (instead of referring as group or family). This helps to have consistency throughout the introduction.

Line 54: What “EVI” means hasn’t been introduced earlier in the introduction.

Line 80: Better to use as “2C protease” instead of “2C protein”, so that it reflects more of its functional role. In addition, as the 2C protease is mentioned for the first time in the introduction, it is better to include a shorter introduction of structural proteins of enteroviruses (or specifically coxsakieviruses) and the known roles of 2C protease in viral replication.

Materials and Methods

Line 88: Use “as described previously” instead of “according to the procedure described previously”

Line 89: Use “are shown in Figure 1.” instead of “given at the Figure 1 below.”

Line 92: It is better if the quality of the pleconaril structure can be improved.

Line 140: Refer to Figure 1 as “compounds 2 through 5 (Figure 1)”

Lines 143 and 150: Use MTT assay instead of using MTT alone

Line 229 and 230: Better to use “MEM” instead of “medium” to be consistent with the rest of the sections.

Line 251: Better to define M±SE as Mean ± Standard Error (M±SE) here, where it appears for the first
time.

Results

Line 256: It will be more accurate if Leucoverdazyls are stated as potent inhibitors of CVB3 instead of EVs, as the data is shown only for CVB3 inhibition.

Line 276: As the IC50 values of different test compounds are compared with that of pleconaril, it is better if the IC50 value of pleconaril is included in the Table 1, to give the reader a better idea of the difference in IC50s.

Line 291: Instead of “we suppose”, it’s better to use a more solid term like “data shows that”

Line 299: Correct “anti-enterovirus activity” as “anti-enteroviral activity”

Lines 312-314: Better to use a term like “1a showed significantly higher antiviral activity in comparison to the reference compound” instead of using “high activity” and “superior”

Line 239: Use “This led us to the assumption that…”

Line 330: “common and important for both enterovirus and coronavirus life cycles”

Line 332: It is better to state the results for CVB4 rather than enteroviruses as it wasn’t tested for many enteroviruses to arrive at that conclusion.

Line 343: mean ± SD, mean ± SE, and mean ± error has been used interchangeably throughout the text. It will be better if one out of the 3 was used throughout.

Line 348: It is better to mention earlier in the paragraph that this thermostability assay was carried out on CVB4. This is the first place in the paragraph where reader knows that this assay was carried out on CVB4. Also, indicate CVB4 within the figure 3 legend.

Line 361: The authors should explain the reason for using different concentrations of compound 1a in different assays. As an example, 10 and 5 ug/ml of compound 1a was used in thermostability and time-of-addition assays, respectively.

Lines 375-381: This information has to be supported by references.

Line 388: The specific virus that was used for the infection should be mentioned within the text.

Line 422: Define/Introduce what is 4PL

Discussion

Lines 476-477: “formation of replication and transcription complexes” can be avoided and having “processes of RNA replication and virion assembly” is enough to provide the reader with the role of replication organelles.

Line 477: Use “host cell’s antiviral defense mechanisms” instead of “antiviral host cell defense mechanisms”

Lines 542-546: Better if moved to the 5. Conclusions section.

Comments on the Quality of English Language

There are certain places that minor editing for grammar is needed. The places within text that can be improved with proper academic terms are suggested under minor revisions.  

Reviewer 2 Report

Comments and Suggestions for Authors

 The authors in the present study identified leucoverdazyls as novel inhibitors of enterovirus replication, demonstrating significant antiviral potential. The lead compound, 1a, showed low cytotoxicity but also high antioxidant and virus-inhibiting activities. These findings underscore the effectiveness of leucoverdazyls against enterovirus replication and highlight the crucial role of 2C proteins in the viral lifecycle. However, I have a minor comment that could be beneficial for the paper.

Figure 1: The figure is not of high quality. Please ensure that you use a high-quality image without pixelation and maintain consistent font size and style.

Table 1: Could you present the data in Table 1 in the form of a graph? This would likely be more effective for the reader.

Table 2: Similar to Table 1, presenting your results in the form of a graph would be more beneficial for the reader.

Figure 3: The image appears pixelated. Please ensure to use a high-quality image.

Figure 5: In your electron microscopy analysis, you focused only on signs of viral replication, such as the formation of vacuoles. Could you also include a comparison of the number of viral particles in mock-treated and treated conditions?

Does compound 1a affect viral genome transcription and viral protein expression?

Comments on the Quality of English Language

The English in this paper could be improved by a native speaker, if possible.

Round 2

Reviewer 1 Report

Comments and Suggestions for Authors

I am happy to see you have used my suggestions for improving the quality of your manuscript. I am happy with the current status of the manuscript. 

Comments on the Quality of English Language

Minor editing of English language required.

Author Response

Thank you very much for taking the time to review the revised variant of the manuscript and for suggesting improvements to the English language of this text. We asked english native speaker to find possible misprints and mistakes and then corrected them according to his advice. The mistakes identified included incorrect usage of articles (lines 51, 188, 232, 307, 320, 355, 458, 459), conjunctions (lines 318), prepositions (92, 274, 493). Also we corrected sentence (line 317), changed fonts in the heading and address lines and in lines 128, 155, 168, 176, 206, 218, 231, 252).  Please find  the corresponding corrections highlighted (in yellow) in the re-submitted file.

Reviewer 2 Report

Comments and Suggestions for Authors

Thank you to the authors for improving the paper. However, I still have one concern about your electron microscopy (EM) images. Is the absence of cellular organelle modification common in your treated condition? Could you provide a statistical comparison graph showing the number of cells with and without this modification?

Author Response

Thank you very much for taking the time to review the revised variant of the manuscript and for suggesting important improvements to the 3.4 section covering TEM study in the present text. To make the results of TEM study more solid we added Figure 5D and changed the description of the figure accordingly (lines 419-425). In Figure 5D we presented the diagram of statistical analysis of cell numbers with and without signs of viral replication between treated and non-treated CVB3 infected groups using X2 (df=1, x2= 39.79, p<0.05). Also we indicated how many images were examined in each group in TEM study (in lines 404-406), and how many out of them were positive for signs of viral replication (lines 407, 411-413). We added lines 267-268 to the Material and Methods section 2.12 (Statistics).